# Functional Fitness of Low-Income Community-Dwelling Older Adults in Amazonian Brazilian

**DOI:** 10.3390/healthcare13202575

**Published:** 2025-10-14

**Authors:** Alex Barreto de Lima, Myrian Abecassis Faber, Miguel Peralta, Helena Vila-Suárez, Duarte Henriques-Neto

**Affiliations:** 1Course of Physical Education, University of Nilton Lins, Manaus 69057-001, Brazil; alex.lima@uniniltonlins.edu.br; 2Skeletal Muscle Assessment Laboratory (LABSIM), Department of Physical Education, School of Technology and Sciences, São Paulo State University (UNESP), Marília 01049-010, Brazil; 3Course of Physical Education, Universidade do Estado do Amazonas, Manaus 69850-010, Brazil; mfaber@uea.edu.br; 4CIPER, Faculdade de Motricidade Humana, Universidade de Lisboa, 1495-751 Lisboa, Portugal; mperalta@fmh.ulisboa.pt; 5ISAMB, Faculdade de Medicina, Universidade de Lisboa, 1649-004 Lisboa, Portugal; 6Faculty of Education and Sports Science, 36005 Pontevedra, Spain; evila@uvigo.gal; 7Healthy Fit Research Group, Galicia Sur Health Research Institute (IIS Galicia Sur), SERGAS-UVIGO, 36213 Vigo, Spain; 8Research Center in Sports Sciences, Health Sciences and Human Development, Maia University, 4475-690 Maia, Portugal

**Keywords:** aging, functional fitness, low-income, public health

## Abstract

**Background:** The functional capacity of older adults is a critical determinant of autonomy and quality of life, particularly in low-income populations from remote regions with limited access to health services. This study aimed to characterize the functional fitness (FF) of community-dwelling older adults in the interior of Amazonas, Brazil, stratified by sex and age group. **Methods:** A cross-sectional study was conducted with 807 older adults (471 females), aged ≥ 60 years, from four municipalities in northern Brazil. The FF was assessed using the Senior Fitness Test (SFT), including measures of strength (30-s Chair Stand Test—CST; 30-s Arm Curl Test—ACT), flexibility (Chair Sit and Reach Test-CSAR, Back Scratch Test-BST), balance/agility (8-Foot Up-and-Go Test—FUG), and aerobic endurance (6-min walk test—6MWT). Descriptive statistics, confidence intervals, and age- and sex-specific percentiles were calculated. **Results:** Results indicated a progressive decline in FF with advancing age. Males outperformed females in strength and endurance tests, whereas females exhibited better flexibility. Notable reductions in performance were observed after age 75, particularly in CST, ACT, FUG, and 6MWT. Overall, the functional profiles of this population were below international norms, especially among females and individuals aged ≥ 80. The prevalence of overweight was 39.3%, with socioeconomic vulnerability affecting over 90% of participants. **Conclusions:** Older adults in low-income, remote Amazonian Brazilian communities demonstrate marked functional decline with ageing, influenced by socioeconomic and environmental constraints. These findings highlight the urgency of implementing accessible, community-based interventions focused on physical activity, strength, mobility, and endurance to support healthy ageing in underserved regions.

## 1. Introduction

Population ageing is a global and irreversible phenomenon that is becoming increasingly evident in developing countries. According to estimates by the World Health Organisation (WHO), by 2030, one in six people worldwide will be aged 60 or older, representing approximately 1.4 billion individuals [1]. The Brazilian Institute of Geography and Statistics (IBGE) estimated that in 2022, there would be around 32.1 million people aged 60 or over, representing 15.1% of the country’s total population [2]. This percentage is projected to reach 25.5% by 2060, totalling approximately 58.2 million older people [2]. This significant growth in Brazil’s older population requires attention to the living and health conditions of this group, especially concerning their autonomy, functionality, and quality of life [3].

In Brazil, population ageing is occurring at an accelerated and uneven pace, with regions such as the North facing specific challenges related to access to health services, infrastructure, and social support [2]. The northern region of Brazil, characterised by its large territory, socio-cultural diversity, and socio-economic inequalities, has a growing older population living in the community [1]. These older people, often exposed to precarious living conditions and with limited access to health care, may have their functionality compromised early on [4].

The aging process is closely related to a progressive decline in physical function, which compromises vital functions essential to maintaining functional independence, social engagement, and quality of life for the older people [5]. This process is characterized by gradual biological changes, such as loss of muscle mass and strength known as sarcopenia, changes in body composition, reduced cardiorespiratory capacity, joint stiffness, and impaired balance, which together have a direct impact on the functionality and autonomy of this population [6]. These changes have a direct impact on functional fitness and, when associated with social and environmental factors, can accelerate the process of dependency and reduce the quality of life of the older people in the region [4].

In this context, functional fitness (FF) is understood as the physical capacity needed to carry out activities of daily living safely, independently and without excessive fatigue [7]. Assessing this capacity in the elderly is of the utmost importance, especially in the context of healthy ageing and containing the costs associated with health care [8]. Functional fitness levels in this population result from a set of interconnected factors, including socioeconomic changes throughout life, housing conditions, sociocultural, ethnic and genetic aspects, as well as factors related to nutrition, geographical region and physical activity habits [9,10]. It is essential to assess how community-based physical activity programs can improve functional fitness, mitigate the risk of falls and fall-related injuries, and, consequently, contribute to improving quality of life in this population [11,12]. Studies show that low functional fitness is associated with a higher risk of hospitalizations, falls, early mortality, and worse mental health indicators [13,14]. In addition, there is evidence that functional fitness has a strong correlation with metabolic health, with physical inactivity and loss of strength being risk factors for diseases such as type 2 diabetes, hypertension, and metabolic syndrome [13].

In northern Brazil, where geographical and social barriers often limit access to health promotion programs, it is even more important to identify the level of functional fitness of the older population living in the community [15]. This diagnosis can support interdisciplinary interventions and public policies that favor active and quality aging [15]. Ultimately, this can inform community health practices aimed at promoting active and healthy aging, thus improving the quality of life for older people in Amazonas and beyond. Thus, this study aims to characterise the FF of older people living in low-income communities in Amazonas (northern Brazil), according to sex and age groups.

## 2. Materials and Methods

### 2.1. Study Design and Participants

A cross-sectional study was carried out in the municipalities of Borba, Novo Aripuanã, Tonantins and Jutaí. Borba is located south of Manaus (capital), 208 kilometres from the capital (latitude—4°23′16″ S, longitude—59°35′38″ O). Novo Aripuanã is located in the mesoregion of Sul Amazonense and microregion of Madeira (latitude—5°08′00″ S, longitude—60°22′30″ O). Tonantins (latitude—2°52′22″ S, longitude—67°48′07″ O) and Jutaí (latitude—2°44′49″ S, longitude—66°46′01″ O) are both located in the Alto Solimões region, 872 and 985 km from the capital, respectively. Access to these municipalities from Manaus and other parts of Brazil is only possible via river routes. For most residents of the Amazon region, boats are the primary mode of transportation. These communities are largely rural, with local economies relying heavily on agriculture and fishing. The Madeira and Solimões Rivers, along with their tributaries, serve as vital lifelines for both livelihood and transportation. The ongoing urbanisation process has led to increased migration from riverside areas to urban centers, driven mainly by the search for better job prospects and income opportunities. The sample consisted of eight hundred and seven older people (male = 336 and female = 471), living in the cities of Borba, Novo Aripuanã, Tonantins, and Jutaí, located in Amazonas, the northern region of Brazil. The sample wasdivided into 6 age groups (60–64, 65–69, 70–74, 75–79, 80–84, and ≥85 years) and comprised 41.6% females and 58.4% males. This study was approved by the Ethics and Research Committee of the State University of Amazonas (UEA) under opinion (CAAE: 43032915.5.0000.5016; 56689416.7.0000.5016; 74055517.9.0000.5016; 56415822.9.0000.5016) by the Declaration of Helsinki [16] and Resolution 466/12 of the National Health Council [17]. Voluntary participants were recruited in basic health units and community centres. Older people living in rural areas were excluded from the study due to difficulties in accessing the assessment location (distance and necessary means of transport). After explanations about the study procedures and risks, all participants signed the free and informed consent form. All assessments were carried out on previously approved ethical principles of the UEA. The following criteria were considered for participant inclusion: (1) older people aged 60 or over living in the community; (2) be independent in carrying out activities of daily living; (3) moderate or high level of cognitive functioning; (4) absence of contraindications for physical exertion (stroke, neurological diseases, unstable chronic conditions); (5) no chest pain and/or angina pectoris and limiting joint pain [18].

### 2.2. Socioeconomic Status

The questionnaire from the Brazilian Association of Research Companies was used to evaluate sociodemographic variables [19]. This questionnaire classifies individuals into five social classes, ranging from class A (with greater purchasing power) to class E (with lower purchasing power), based on possession of consumer goods, education of the head of the family, and access to public services.

### 2.3. Anthropometric Measurements

The weight and height were measured using a mechanical scale with an attached stadiometer (Welmy, São Paulo City, Brazil). The Body Mass Index (BMI) was calculated based on these measurements. The following categories were adapted: underweight < 18.5 kg/m^2^; normal 18.5–24.9 kg/m^2^; overweight 25–29.9 kg/m^2^; obesity > 30 kg/m^2^, in accordance with criteria provided by WHO [20].

### 2.4. Functional Fitness Tests

The FF was assessed using the Senior Fitness Test (SFT) battery, developed and validated by Rikli and Jones (2013) to identify older adults at risk of functional loss at an early stage [7]. The SFT battery is practical and widely used to assess the FF of the older adult population [21]. It includes seven tests that measure five dimensions of FF that are strongly associated with functional mobility in independent older adults. The tests are: the 30-s chair stand test (CST) for lower body strength, the 30-s arm curl test (ACT) for upper body strength, the back scratch test (BST) to measure upper limb flexibility, the chair sit and reach test (CSAR) to assess lower limb flexibility, the 8-foot stand and walk test (FUG) to measure motor agility and dynamic balance, and the 6-min walk test (6MWT) to assess aerobic endurance. All the tests were carried out strictly following the instructions in the SFT Manual [18]. To avoid the effects of fatigue contamination, the functional fitness tests were carried out in the following order and sequence:

**30-s Chair Stand Test (CST)**—assessment of lower limb strength. The participant sits on a chair of appropriate height (43 cm high). The arms are crossed at chest height. The task is to perform as many correct lifts (repetitions) of the chair as possible in 30 s.

**30-s Arm Curl Test (ACT)**—upper body strength assessment. The participant receives weight in their dominant hand. The weight is 2.27 kg (5 lb) for females and 3.63 kg (8 lb) for men. The task consists of performing as many correct forearm flexions as possible in 30 s.

**Chair Sit and Reach Test (CSAR)**—assessment of the flexibility of the lower body. The participant sits on the edge of the chair. One leg is stretched out with the heel resting on the floor and the toes pointing upwards. The other leg is bent with the whole foot on the floor. The task is to reach out with your hands towards your toes as far as possible. The distance between the middle finger of the hand and the toes is measured to an accuracy of 0.5 cm.

**Back Scratch Test (BST)**—assessment of the flexibility of the upper body. The person stands and places one hand on their back, from top to bottom, and the other from bottom to top. They try to bring the fingers of both hands together. The distance between the middle fingers of the two hands is measured to the nearest 0.5 cm.

**8-Foot Up-and-Go Test (FUG)**—agility and dynamic balance assessment. The person is sitting on a chair with a bollard in front of them, at a distance of 2.44 m. At the signal, the person gets up, goes to the pole, goes around it, returns to the chair, and sits down. The task must be carried out as quickly as possible. The result is the time taken to complete the task.

**6-min walk test (6MWT)**—aerobic endurance rating. The person being examined walks quickly along a rectangle with total dimensions of 45.72 m in 6 min. The result is the total distance covered by the person during that time (6 min).

The sequence of tests was defined based on the physiological principles of bioenergetics. Thus, participants were asked to perform tests that alternate muscle groups (i.e., lower limbs-upper limbs) and intensity levels (i.e., high intensity-moderate or light intensity). The protocols that recruited the largest muscle groups and were of higher intensity were performed first. Protocols such as the 6MWT, which are of lower intensity, were performed at the end of the protocols. A minimum recovery period of 2 min was also allowed between protocols (transition period between assessment spaces).

### 2.5. Statistical Analysis

All statistical analyses were performed using IBM SPSS Statistics (version 28) for Windows (SPSS Inc., Chicago, IL, USA) with a significant level of 5%. The data are presented stratified by age, reported as mean and standard deviation and confidence interval (CI) (95%) for continuous variables and as frequencies and percentages for categorical variables. The Levene’s test was used to analyse the variances. Normality was not formally tested, as the central limit theorem supports the approximation of the sampling distribution to normality with sufficiently large sample sizes. Differences between sexes were examined using independent-samples *t* tests for continuous variables and chi-square tests for categorical variables. When chi-square tests were significant, post hoc pairwise comparisons with Bonferroni correction were applied to adjust for type I error inflation. Comparisons between age groups within each sex were performed using one-way analysis of variance (ANOVA) for all functional fitness variables. The 10th, 25th, 50th, 75th, and 90th percentiles were chosen as the normative reference points for each 5-year age group (60–64, 65–69, 70–74, 75–79, 80–84, and ≥85).

## 3. Results

Table 1 shows the characteristics of the participants. In the overall sample (n = 807), the mean age was 72.2 years (95% CI: 71.6–72.8), with no significant differences between male and females across age groups. Regarding socioeconomic status, the majority of participants were classified in the lowest categories (D+E), with no sex-related differences. In terms of body composition, males presented significantly higher body mass (67.9 kg vs. 60.0 kg; *p* < 0.001) and height (158.9 cm vs. 149.6 cm; *p* < 0.001) compared with females, although BMI did not differ between sexes. The FF analysis revealed consistent sex-related differences: males outperformed females in the CST (11.3 vs. 10.3 repetitions; *p* < 0.001), ACT (11.7 vs. 10.8 repetitions; *p* < 0.001), and 6MWT (443.8 m vs. 384.0 m; *p* < 0.001). Conversely, females performed better in flexibility measures, with superior results in the CSAR (4.7 vs. 3.2 cm; *p* < 0.001) and the BST (−13.8 vs. −18.5 cm; *p* < 0.001). Additionally, males demonstrated faster performance in the FUG (8.1 vs. 8.9 s; *p* < 0.001). These findings indicate a clear pattern of sex-related differences in body composition and FF, with males showing advantages in strength, mobility, and aerobic endurance, while females exhibit higher flexibility levels.

In Table 2, FF values were stratified by age group and sex. In males, BMI remained relatively stable across age groups, ranging from 27.3 kg/m^2^ (95% CI: 26.4–28.1) in the 65–69 age group to 26.0 kg/m^2^ (95% CI: 24.5–27.6) in individuals aged over 85 years. Performance on the CST showed a gradual decline with age, from 12.1 repetitions (95% CI: 11.3–12.9) at 65–69 years to 9.9 repetitions (95% CI: 8.9–10.9) in those over 85. Similarly, results from the ACT decreased from 12.7 repetitions (95% CI: 11.9–13.5) to 9.9 repetitions (95% CI: 8.5–11.3) across the same age span. The CSAR scores demonstrated a marked decline with advancing age, from 5.6 cm (95% CI: 3.1–8.2) at ages 65–69 to 0.9 cm (95% CI: –3.2 to 5.2) in those over 85. The BST showed worsening decline from –13.9 cm (95% CI: –17.6 to –10.2) at 60–64 years to –23.4 cm (95% CI: –27.3 to –19.5) in the oldest group. The FUG scores revealed an increase in completion time from 7.2 s (95% CI: 6.7–7.7) at 65–69 years to 9.7 s (95% CI: 8.8–10.6) at over 85.

The 6MWT decreased significantly with age, from 468.2 m (95% CI: 447.9–488.6) at 65–69 years to 374.1 m (95% CI: 349.3–398.8) in those over 85. Among females, BMI showed a slight decrease with age, from 27.9 kg/m^2^ (95% CI: 26.9–28.9) at 60–64 years to 24.4 kg/m^2^ (95% CI: 23.2–25.5) at 85+. CST and ACT results mirrored those of males, declining with age. CST declined from 10.8 repetitions (95% CI: 10.2–11.4) at 65–69 years to 7.6 repetitions (95% CI: 6.8–8.4) in the oldest group, while ACT decreased from 11.7 repetitions (95% CI: 11.1–12.4) to 7.9 repetitions (95% CI: 6.9–9.1). The CSAR results decreased from 7.0 cm (95% CI: 4.5–9.4) at 60–64 years to 0.3 cm (95% CI: –2.8 to 3.4) at over 85. The BST results remained negative across all age groups, indicating limited flexibility, with the poorest result observed in those over 85 years (–17.1 cm; 95% CI: –20.3 to –13.9). The FUG times increased with age, rising from 8.2 s (95% CI: 7.7–8.7) at 60–64 years to 11.7 s (95% CI: 10.5–12.9) at over 85. Finally, 6MWT performance decreased consistently with age, from 420.7 m (95% CI: 400.7–440.7) to 307.7 m (95% CI: 274.0–341.4) in those aged over 85. These findings indicate a clear pattern of sex-related differences in body composition and functional fitness, with male showing advantages in strength, mobility, and aerobic endurance, while female exhibit superior flexibility.

Table 3 presents norm values, calculated using the 10th, 25th, 50th, 75th, and 90th percentiles, for individual SFT tests within each 5-year age group in males and females. The median BMI values (50th) remained relatively stable between the age groups, with a slight reduction in the older groups. For males, the values ranged from 25.1 kg/m^2^ (≥85 years) to 26.8 kg/m^2^ (65–69 years), while for females, the values ranged from 23.9 kg/m^2^ (≥85 years) to 28.0 kg/m^2^ (60–64 years). These results indicate a tendency for body mass to decrease with age, especially in females. Also, FF scores showed a clear age-related decline, more pronounced after 75 years. In the CST, the results show that there is a progressive reduction in the number of repetitions with increasing age. In males, the median varied from 12 repetitions (65–69 years) to 10 (≥85 years), while in females, the decrease was more marked, from 11 (60–64 years) to 8 (≥85 years), with markedly lower percentiles in the latter age group. Upper limb strength as assessed by the 30-s push-up test (ACT) decreased with age, especially among older females. Males had medians ranging from 13 repetitions (65–69 years) to 10 (≥85 years). Among females, the medians fell from 11 (65–69 years) to 9 repetitions in the ≥85 years group, with the 10th percentile in this group being 1 repetition. The CSAR and BST were used to assess flexibility. The CSAR results show that there is greater variability between the sexes. Females performed better, with median values increasing at older ages (for example, 10 cm in the 80–84 age group), while males maintained median values close to 0 cm or even negative, indicating less flexibility. In the BST, both sexes showed a decline in flexibility with increasing age. The medians ranged from −13.5 cm (60–64 years) to −24.0 cm (≥85 years) in males and from −12.0 cm to −20.5 cm in females, reflecting less shoulder mobility in the older age groups. When assessing agility and balance using the FUG, it was possible to see an increase in times with advancing age. Males went from 8.1 s (60–64 years) to 9.5 s (≥85 years), while females had a more significant increase: from 7.4 s to 11.1 s in the same age groups. In the six-minute walk test (6MWT), the distance covered showed a downward trend with age. Males showed greater distances in all percentiles, with medians from 473 m (60–64 years) to 360 m (≥85 years). Among females, the medians fell from 400 m to 306 m. There was a low value at the 10th percentile for females ≥ 85 years (165 m).

## 4. Discussion

This study aimed to present descriptive values of the functional fitness levels of low-income community-dwelling elderly people in Amazonas (northern Brazil), according to sex and age groups. The FF is essential for general health and well-being and is crucial for carrying out the daily activities of the human body. It is directly related to fundamental movement skills such as walking, running, squatting, reaching, and manipulating objects [22]. On the other hand, reduced functional fitness can harm daily tasks, such as housework and leisure-time physical activities. Both sexes showed a gradual decline in physical performance across aging, but the trajectory differed markedly between male and female.

While men preserved strength and aerobic capacity until approximately 75 years, female exhibited earlier and more linear reductions, particularly in lower-limb strength and walking endurance. Functional decline was most pronounced after 80 years in both groups, yet females experienced disproportionately greater losses in mobility and gait efficiency. Flexibility followed sex-specific trends, with male showing larger deficits in shoulder range of motion and female presenting greater impairments in agility-based tasks. Interestingly, BMI remained relatively stable until very advanced age, suggesting that functional deterioration precedes changes in body composition.

Firstly, ageing is related to a decline in functional fitness, which is particularly evident in low-income elderly people, who may face additional difficulties in preserving their physical health [23]. Kang et al. (2021) point out that functional fitness is fundamental for maintaining independence in life, but tends to decrease with advancing age, showing significant differences between the elderly, depending on their socioeconomic status [24]. In corroboration, Hayajneh and Rababa (2022) found that poverty can lead to frailty, which drastically affects the ability of the elderly to engage in physical activities and social interactions necessary to maintain functional fitness [25]. The results of this study show a clear trend of decline in functional fitness with advancing age among the elderly living in low-income communities in northern Brazil. Both male and female showed a significant reduction in strength test scores (CST and ACT), flexibility (CSAR and BST), agility and dynamic balance (FUG) and aerobic endurance (6MWT) from the age of 70, with greater accentuation after the age of 80. These findings are in line with international literature, which points to the loss of functional fitness as a common factor in the ageing process [26,27,28,29].

When compared to the data from Cossio-Bolaños et al. (2025) in elderly people from Chile, it can be seen that the elderly from the Amazon showed inferior performance in almost all the tests, especially in the 6-min walk test (6MWT), in which the Chilean male aged 60–64 covered an average of 580 m, while the Brazilians reached only 463.6 m [30]. This discrepancy of more than 100 m reflects inequalities in access to health care and more sedentary lifestyle habits among socially vulnerable populations [30].

Studies such as those by Rikli and Jones (1999) have already shown that muscle strength and aerobic capacity show progressive declines with advancing age, directly impacting the ability to carry out activities of daily living [7]. The values found for CST and ACT in this study are similar to those presented in several studies conducted in different parts of the world in older people [26,27]. In Portugal, Rodrigues et al. (2024) evaluated elderly people from the Leiria community and found averages of 14.5 repetitions in the sit-to-stand test (CST) and 17.1 repetitions in the arm strength test (ACT) for both sexes between 60 and 69 years old. In the present study, Amazonian males presented similar values (CST = 12.0; ACT = 11.6), but females were below (CST = 10.9; ACT = 11.0), suggesting disparities related to socioeconomic status, level of education, and regular practice of physical activity [31].

Based on the reference values proposed by Rikli and Jones (1999), widely used in North American populations, it is noted that the elderly Brazilians studied are close to or slightly below the 25th percentile for most tests, especially in those over 80 years of age [7]. These data reinforce the negative impact of accelerated ageing in regions with less social support and health promotion infrastructure. When comparing our data with those of Wang et al. (2024), who evaluated 3332 elderly people in rural southern Taiwan, it is observed that Brazilian elderly people presented lower performance, especially in strength and endurance measures. While Taiwanese males aged 60 to 64 achieved an average of 15.7 repetitions in the sit-to-stand test (CST), Brazilians obtained an average of 12.0 repetitions [32]. This may be related to lower access to regular physical activity programs in the Amazon population. Similarly, Kang et al.’s (2021) longitudinal study of elderly South Koreans demonstrated a sharp decline in strength, flexibility, and endurance over four years [24]. In the present study, this pattern was also evidenced in a cross-sectional manner, with clear drops in the average values in all age groups. Also, Korean authors highlight the importance of regular physical exercise as a protective factor, something that can be explored as a public policy in Brazilian reality. Furthermore, the study by Abreu et al. (2021), carried out in Brazil, identified that females have worse functional performance in tests such as walking speed and lower limb strength, which was also observed in the present sample [33].

Anthropometric and hormonal differences between the sexes may justify this inequality, reinforcing the need for sex-differentiated strategies. Regarding flexibility, data from the Amazon population also indicate greater deficits. The average values of the seated reach test (CSAR) and the backhand touch test (BST) were below those observed in European countries such as Spain, Poland, and China, as reported by international comparative studies [30,34]. This implies a greater functional risk, especially for activities such as putting on shoes or reaching for high objects, which can compromise independence. Furthermore, the present study highlights a high prevalence of overweight and obesity (62.5%), a factor that can aggravate functional decline. As evidenced by Rodrigues et al. (2024), older people with normal BMI showed significantly better performance in all physical tests compared to those who were overweight or obese, which was also confirmed in this study [31].

The decline in functional fitness is consistently observed across aerobic capacity, flexibility, and muscle strength; however, this deterioration is more accentuated in females, particularly with respect to muscle strength. Data from this study revealed significant sex-based differences favouring males, except for flexibility. The BMI did not differ significantly between sexes or across age groups. These findings suggest that, overall, participants do not experience substantial reductions in body weight with ageing; rather, they demonstrate a progressive decline in muscle strength and aerobic endurance, a trend that becomes particularly evident after the age of 70.

This phenomenon can be explained through both biological and sociocultural mechanisms. From a biological perspective, females present lower absolute muscle mass and bone mineral density across the lifespan, even after adjusting for anthropometric variables such as height and body weight. With advancing age, the processes of sarcopenia, dynapenia and powerpenia manifest more markedly in females, a pattern largely attributable to reduced production of anabolic hormones (e.g., testosterone) and the abrupt decline in estrogen following menopause, which accelerates the deterioration of muscle mass [35,36,37].

On a sociocultural and economic level, gender inequalities exert a potential critical influence. Several studies indicate that socioeconomic status can significantly influence levels of functional fitness and mental health outcomes in this population [38]. In low-income contexts, females are disproportionately exposed to cumulative burdens of unpaid physical labour, reduced opportunities for structured exercise and leisure-time physical activity, and dietary restrictions that compromise protein and micronutrient intake necessary for preserving muscle mass [39,40,41]. Additionally, lower educational attainment and limited health literacy, common in such environments, restrict access to preventive and rehabilitative health care [42]. Accumulated social vulnerability further amplifies women’s exposure to disability-related risk factors, including malnutrition, informal work overload, and limited availability of assistive technologies. Consequently, the greater functional decline observed in older women, particularly in low-income societies, should be understood not solely as a biological consequence of ageing but rather as the outcome of complex interactions between gender-based inequalities, socioeconomic disadvantage, and environmental determinants [43].

### 4.1. Practical Implications

The assessment of functional fitness in older adults has demonstrated that it is an excellent tool for evaluating and monitoring the health profile of older adults, especially in low-income populations with limited access and a high degree of geographical isolation, as is the case with the sample in this study. Additionally, the findings of this study reinforce the urgent need for public policies aimed at promoting physical activity and functional rehabilitation in older individuals in the Amazon region. Thus, promoting accessible and continuous community programmes can mitigate the harmful effects of ageing and ensure greater functional independence, especially in people over 70 and older females of all age groups.

Overall, the results reinforce the need for specific intervention strategies to maintain functional fitness in low-income older adults. Community programmes focusing on muscle strength, mobility, and aerobic capacity should be implemented to delay functional decline and promote healthy ageing.

### 4.2. Strengths and Limitations

This study presents several methodological strengths that warrant consideration. Notably, the use of standardised and validated physical fitness assessments in an understudied older adult population enhances the reliability of intergroup comparisons and facilitates benchmarking with international datasets. The inclusion of both male and female participants across a broad age spectrum further strengthens the external validity of the findings, particularly concerning older adults residing in remote regions of the Brazilian Amazon. These areas are characterised by geographic isolation and limited socioeconomic resources, posing substantial challenges to the implementation of large-scale scientific investigations. Moreover, the results contribute valuable knowledge regarding the functional profile of this population, thereby informing the formulation of context-specific public health interventions and policies. Despite these strengths, certain limitations should be acknowledged. First, the sample size was insufficient to provide a representative distribution of older adults across all age strata, rural areas and municipalities within the interior of the Amazonas Brazilian state. Second, the absence of data on participants’ physical activity levels constrains the interpretability of the FF outcomes. Future research should incorporate assessments of body composition and physical activity to provide a more comprehensive understanding of the determinants of functional capacity in this population.

## 5. Conclusions

The results of this study show consistent differences between the sexes and a clear trend of functional decline with age. Males showed superior performance in most strength and endurance tests, while females showed better flexibility (CSAR and BST). The decline is particularly evident after the age of 75, suggesting the importance of promoting physical activity and healthy lifestyle strategies to promote functionality and health in older adults.

## Figures and Tables

**Table 1 healthcare-13-02575-t001:** Descriptive characteristics of participants as mean ± standard deviation.

Variables	Mean—(95% CI)				
	All (n = 807)	Male (n = 336)	Female (n = 471)	*t/x^2^*	*p-Value*
All age groups	72.2 (71.6 to 72.8)	72.4 (71.6 to 73.3)	72.1 (71.2 to 72.9)	0.565	0.286
**Age group (years)**					
60–64	62.1 (61,9 to 62.4)	62.3 (61.9 to 62.6)	62.1 (61.8 to 62.4)	0.755	0226
65–69	66.9 (66.7 to 67.1)	66.9 (66.6 to 67.2)	66.9 (66.7 to 67.2)	−0.0273	0.392
70–74	71.9 (71.6 to 72.1)	72.0 (71.7 to 72.3)	71.7 (71.4 to 71.9)	1.558	0.061
75–79	76.8 (76.6 to 77.1)	77.1 (76.7 to 77.5)	76.7 (76.3 to 77.0)	1.543	0.063
80–84	81.8 (81.4 to 82.1)	81.8 (81.3 to 82.3)	81.8 (81.3 to 82.3)	−0.60	0.476
≥85	89.9 (88.9 to 90.9)	89.1 (87.4 to 90.7)	90.4 (89.1 to 91.7)	−1.235	0.110
**Socioeconomic level, n (%)**					
A+B (best condition)	0.5 (0.0 to 0.01)	0.3 (0.0 to 0.9)	0.6 (0.0 to 1.4)	0.409	0.783
C	6.9 (0.5 to 09)	7.1 (4.4 to 9.9)	6.8 (4.5 to 9.1)		
D+E (worst condition)	92.6 (91 to 94)	92.6 (90 to 95)	92.6 (90 to 95)		
** *Body composition* **					
Body Mass, kg	63.3 (62.4 to 64.2)	67.9 (66.7 to 69.1)	60.0 (58.0 to 61.1)	9.319	<0.001
Body Height, cm	153.5 (152.9 to 154.1)	158.9 (158.2 to 159.8)	149.6 (149.0 to 150.2)	18.419	<0.001
BMI, kg/m^2^	26.8 (26.5 to 27.1)	26.9 (26.4 to 27.3)	26.7 (26.3 to 27.1)	0.422	0.337
** *Functional Fitness* **					
30-CST, n	10.7 (10.5 to 10.9)	11.3 (10.9 to 11.6)	10.3 (10.0 to 10.3)	4.276	<0.001
ACT, n	11.2 (10.8 to 11.4)	11.7 (11.3 to 12.1)	10.8 (10.4 to 11.1)	3.183	<0.001
CSAR, cm	4.1 (3.3 to 4.8)	3.2 (2.1 to 4.2)	4.7 (3.7 to 5.7)	−1.915	<0.001
BST, cm	−15.8 (−16.7 to −14.8)	−18.5 (−19.9 to −17.0)	−13.8 (−15.0 to −12.7)	−5.013	<0.001
FUG, sec	8.6 (8.3 to 8.8)	8.1 (7.6 to 8.5)	8.9 (8.6 to 9.2)	−3.379	<0.001
6MWT, m	408.9 (401.4 to 416.4)	443.8 (431.7 to 455.9)	384.0 (375.1 to 392.9)	7.813	<0.001

**Legend**: CI, confidence interval; BMI, body mass index; 30-CST, 30-s chair stand test; ACT, 30-s arm curl test; CSAR, chair sit-and-reach test; BST, back scratch test; FUG, 8-foot up-and-go test; 6MWT, 6-min walk test; values expressed as Mean and 95% CI.

**Table 2 healthcare-13-02575-t002:** Functional Fitness by age group in Amazonas, Brazil.

		Age Group				
	60–64 yrs	65–69 yrs	70–74 yrs	75–79 yrs	80–84 yrs	≥85 yrs
	Mean—(95% CI)	Mean—(95% CI)	Mean—(95% CI)	Mean—(95% CI)	Mean—(95% CI)	Mean—(95% CI)
**Male**						
BMI	27.1 (25.9 to 28.3)	27.3 (26.4 to 28.1)	26.7 (25.9 to 27.5)	27.1 (25.6 to 28.7)	26.2 (25.2 to 27.1)	26.0 (24.5 to 27.6)
CST (n) ^i^	12.0 (11.3 to 12.8)	12.1 (11.3 to 12.9)	11.2 (10.5 to 12.0)	10.6 (9.7 to 11.6)	10.4 (9.4 to 11.4)	9.9 (8.9 to 10.9)
ACT (n) ^i^	11.6 (10.4 to 12.7)	12.7 (11.9 to 13.5)	12.0 (11.0 to 13.0)	11.6 (10.5 to 12.7)	10.6 (9.4 to 11.8)	9.9 (8.5 to 11.3)
CSAR (cm)	4.0 (1.4 to 6.6)	5.6 (3.1 to 8.2)	2.9 (0.6 to 5.2)	1.7 (−0.5 to 4.0)	1.2 (−1.4 to 3.8)	0.9 (−3.2 to 5.2)
BST (cm) ^d,e^	−13.9 (−17.6 to −10.2)	−17.1 (−20.1 to −14.1)	−19.2 (−22.1 to −16.2)	−18.3 (−22.1 to −14.4)	−23.9 (−28.1 to −19.8)	−23.4 (−27.3 to −19.5)
FUG (seg) ^h,i^	7.7 (5.8 to 9.6)	7.2 (6.7 to 7.7)	7.9 (7.3 to 8.4)	7.9 (7.2 to 8.7)	9.6 (8.8 to 10.3)	9.7 (8.8 to 10.6)
6MWT (m) ^e,i,l^	463.6 (438.1 to 489.1)	468.2 (447.9 to 488.6)	454.5 (429.6 to 479.3)	428.1 (388.0 to 468.3)	415.0 (370.9 to 459.2)	374.1 (349.3 to 398.8)
**Female**						
BMI ^e,i,^	27.9 (26.9 to 28.9)	26.9 (26.1 to 27.5)	26.9 (25.7 to 27.9)	26.7 (25.6 to 27.9)	26.4 (25.1 to 27.7)	24.4 (23.2 to 25.5)
CST (n) ^e,i,l,n^	10.7 (10.3 to 11.4)	10.8 (10.2 to 11.4)	10.7 (10.0 to 11.4)	10.4 (9.7 to 11.1)	9.4 (8.6 to 10.2)	7.6 (6.8 to 8.4)
ACT (n) ^e,i,l,n^	10.9 (10.2 to 11.8)	11.7 (11.1 to 12.4)	11.3 (10.4 to 12.2)	10.5 (9.4 to 11.6)	10.0 (8.7 to 11.4)	7.9 (6.9 to 9.1)
CSAR (cm) ^e^	7.0 (4.5 to 9.4)	5.3 (3.3 to 7.3)	4.4 (2.5 to 6.4)	5.4 (3.2 to 7.5)	1.4 (−2.6 to 5.4)	0.3 (−2.8 to 3.4)
BST (cm) ^l^	−12.2 (−14.8 to −9.5)	−14.7 (−16.9 to −12.5)	−10.3 (−12.9 to −7.7)	−16.4 (−19.4 to −13.3)	−12.9 (−17.3 to 08.4)	−17.1 (−20.3 to −13.9)
FUG (seg) ^e,i,l,r^	8.2 (7.7 to 8.7)	8.3 (7.8 to 8.8)	8.2 (7.7 to 8.8)	9.2 (8.6 to 9.9)	9.7 (8.6 to 10.7)	11.7 (10.5 to 12.9)
6MWT (m) ^c,e,i,l,n,r^	420.7 (400.7 to 440.7)	391.9 (376.3 to 407.6)	391.6 (373.3 to 409.9)	372.5 (353.4 to 391.5)	376.04 (342.9 to 408.1)	307.7 (274.0 to 341.4)

Legend: SD, standard deviation; CST, 30-s chair stand test. ACT, 30-s arm curl test. CSAR, chair sit-and-reach test. BST, back scratch test. FUG, foot up-and go test. 6MWT, 6-min walk test. (c) *p* ≤ 0.05, significant difference between 60–64 and 75–80; (d) *p* ≤ 0.05, significant difference between 60–64 and 80–84; (e) *p* ≤ 0.05, significant difference between 60–64 and ≥85; (h) *p* ≤ 0.05, significant difference between 65–69 and 80–84; (i) *p* ≤ 0.05, significant difference between 65–69 and ≥85; (l) *p* ≤ 0.05, significant difference between 70–74 and ≥85; (n) *p* ≤ 0.05, significant difference between 75–79 and ≥85; (r) *p* ≤ 0.05, significant difference between 80–84 and ≥85.

**Table 3 healthcare-13-02575-t003:** Functional fitness test percentiles in older adults by age group in Amazonas, Brazil.

	Male (n = 336)						Female (n = 471)					
	n	10th	25th	50th	75th	90th	n	10th	25th	50th	75th	90th
BMI												
60–64	60	21.1	23.8	25.9	31.3	34.2	94	22.5	24.2	28.0	30.5	34.4
65–69	79	22.5	24.2	26.8	29.7	32.8	131	21.6	23.7	26.0	29.7	32.4
70–74	77	22.9	23.9	26.5	29.1	31.0	82	21.8	23.6	26.3	28.7	34.1
75–79	53	21.7	23.7	26.5	29.3	31.7	75	21.4	23.2	26.3	29.9	33.8
80–84	38	22.4	24.1	26.5	28.0	31.2	37	21.6	23.3	25.9	29.3	31.7
≥85	29	21.0	23.3	25.1	28.6	33.2	52	19.9	20.9	23.9	26.9	30.3
CST (rep)												
60–64	60	9.0	10.0	11.0	14.0	15.9	94	7.5	10.0	11.0	12.0	13.0
65–69	79	9.0	10.0	12.0	13.0	17.0	131	7.0	9.0	10.0	13.0	15.0
70–74	77	7.0	9.0	11.0	13.0	15.0	82	7.0	9.0	11.0	12.3	14.7
75–79	53	8.0	9.0	10.0	12.5	14.0	75	7.0	9.0	10.0	12.0	15.0
80–84	38	6.0	9.8	10.0	12.0	15.0	37	5.8	8.0	10.0	11.0	12.0
≥85	29	6.0	8.0	10.0	12.0	14.0	52	0.9	7.0	8.0	10.0	10.0
ACT (rep)												
60–64	60	6.0	9.0	10.5	14.8	18.0	94	7.0	8.8	10.0	13.0	16.0
65–69	79	9.0	10.0	13.0	15.0	16.0	131	8.0	9.0	11.0	14.0	17.0
70–74	77	8.0	9.0	11.0	14.0	17.2	82	7.0	9.0	10.0	14.0	17.0
75–79	53	6.4	10.0	12.0	15.0	16.0	75	3.0	8.0	10.0	14.0	16.4
80–84	38	6.9	8.0	10.5	13.3	15.0	37	5.8	8.0	10.0	11.0	12.0
≥85	29	7.0	8.0	10.0	12.0	15.0	52	0.0	6.0	9.0	10.0	12.0
CSAR (cm)												
60–64	60	−7.9	0.0	5.0	10.8	15.0	94	−0.5	0.0	4.0	12.3	23.0
65–69	79	−8.0	0.0	5.0	12.5	20.0	131	−6.6	0.0	3.0	12.0	20.8
70–74	77	−4.0	0.0	3.0	9.0	13.6	82	0.0	0.0	3.0	10.0	17.0
75–79	53	−11.0	0.0	3.0	7.0	10.0	75	0.0	0.0	3.0	10.0	17.8
80–84	38	−6.3	0.0	0.0	4.0	12.1	37	5.0	8.0	10.0	12.5	15.4
≥85	29	−12.0	−6.5	2.0	7.5	12.0	52	−14.0	−4.1	0.0	4.7	12.4
BST (cm)												
60–64	60	−30.0	−27.8	−13.5	0.0	2.9	94	−30.0	−22.0	−12.0	0.0	2.0
65–69	79	−32.0	−29.0	−18.0	−9.0	2.0	131	−30.0	−25.0	−15.0	−5.0	0.8
70–74	77	−32.0	−30.0	−21.0	−14.0	−0.8	82	−29.1	−21.0	−8.3	0.0	3.0
75–79	53	−31.2	−29.5	−23.0	−9.0	2.6	75	−31.4	−26.0	−17.0	−10.0	0.4
80–84	38	−38.1	−30.5	−26.0	−19.0	−2.6	37	−31.8	−23.0	−13.0	0.0	2.2
≥85	29	−36.0	−30.0	−24.0	−17.0	−8.0	52	−30.0	−25.0	−20.5	−10.3	0.0
FUG (seg)												
60–64	60	5.3	6.2	8.1	10.5	12.5	94	5.8	6.5	7.4	9.5	12.1
65–69	79	4.8	5.8	6.7	8.2	10.6	131	5.5	6.4	7.8	9.9	11.9
70–74	77	5.2	6.3	7.3	9.0	12.0	82	5.8	6.6	7.8	9.8	10.9
75–79	53	5.2	6.7	7.5	9.8	10.5	75	5.7	7.2	8.4	10.7	13.7
80–84	38	6.3	7.5	9.4	11.4	12.0	37	6.3	7.1	9.2	10.8	15.1
≥85	29	7.0	8.0	9.5	10.9	13.1	52	8.0	9.8	11.1	14.0	17.3
6MWT (m)												
60–64	60	313.1	385.0	473.0	505.8	600.0	94	320.0	355.8	400.0	480.0	550.0
65–69	79	380.0	400.0	460.0	512.0	600.0	131	286.0	344.0	400.0	454.0	498.0
70–74	77	345.0	392.5	450.0	499.0	600.0	82	293.0	357.8	385.0	431.1	500.0
75–79	53	310.4	359.0	422.0	494.0	588.2	75	259.0	316.0	375.0	433.9	480.0
80–84	38	290.8	331.3	384.0	480.0	532.2	37	250.0	300.0	380.0	421.0	458.3
≥85	29	300.0	326.0	360.0	400.0	450.0	52	165.0	230.0	306.0	393.0	446.1

Legend: BMI, Body Mass Index; CST, 30 s chair stand test.; ACT, 30 s arm curl test; CSAR, chair sit-and-reach test; BST, back scratch test; FUG, foot up-and go test; 6MWT, 6-min walk test.

## Data Availability

Data are available upon request to the corresponding author and with a plausible and scientific justification.

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
