# Peer review of "Functional Fitness of Low-Income Community-Dwelling Older Adults in Amazonian Brazilian"

_healthcare, 2025, doi:10.3390/healthcare13202575_

Round 1
Reviewer 1 Report
Comments and Suggestions for Authors
In the manuscript titled Functional Fitness of Low-income Community-Dwelling Older Adults in Amazonian Brazilian, the authors conducted a cross-sectional study involving 807 low-income community-dwelling older adults in the Amazon region. They assessed functional fitness using the Senior Fitness Test, providing detailed data on the functional fitness of this population and revealing the impact of socioeconomic and environmental factors on the functional abilities of older adults. However, the paper has some shortcomings that should be addressed:
Major comments:
- Line125: The inclusion criteria state that older adults with "moderate or high level of cognitive functioning" were included. How was the cognitive function level of participants defined? It is recommended to specify the assessment tools and cutoff values used.
- Line 153: The rationale for why the test order could avoid fatigue was not sufficiently explained. For example, was the order arranged from low-intensity to high-intensity tests?
- The effect of rural sample exclusion on the representativeness of the results should be explained more fully.
- Both the abstract and results mention that females performed better in certain aspects, but the reasons for gender differences are not sufficiently discussed. Please provide additional elaboration.
- The discussion of intervention recommendations is too general and should be more specific and consider feasibility.
- The discussion section should be logically structured with clear and natural paragraph divisions.
Minor comments:
- Further examine the language of the article: Line 188: "Males were significantly taller and heavier compared to the females of their females" ; Line 114: "under opinion".
- Line 258: the citation "Kang et al." should be followed by a reference number, rather than placing the number at the end of the sentence.
- Please ensure consistency in the reference citation format (e.g., lines 270, 287, and 297).
- Table 1: Please carefully verify the accuracy of the data. For example: "0.5 (0.0 to 0.01)", "6.9 (0.5 to 09)", and "92.6 (91 to 94" seems inconsistent or has an incomplete parenthesis.
Author Response
Dear Reviewer,
On behalf of the manuscript authors, we appreciate all comments from the reviewers. The authors believe that the comments have significantly improved the quality of the manuscript. For better analysis by the reviewers, we have responded individually to all comments.
Available for further information.
Kind Regards
Duarte Henriques-Neto
Reviewer 1
In the manuscript titled Functional Fitness of Low-income Community-Dwelling Older Adults in Amazonian Brazilian, the authors conducted a cross-sectional study involving 807 low-income community-dwelling older adults in the Amazon region. They assessed functional fitness using the Senior Fitness Test, providing detailed data on the functional fitness of this population and revealing the impact of socioeconomic and environmental factors on the functional abilities of older adults. However, the paper has some shortcomings that should be addressed:
Major comments:
- Line125: The inclusion criteria state that older adults with "moderate or high level of cognitive functioning" were included. How was the cognitive function level of participants defined? It is recommended to specify the assessment tools and cutoff values used. Response: The inclusion criteria were defined by the research team and approved by the ethics committee of the State University of Amazonas-Brazil, with the code identified in the methods section. Information on some criteria is clinical, sensitive, and depends on medical opinion. This verification is carried out by those responsible for the community centres that identify participants who meet the inclusion criteria for this study, since they are the ones who have the personal and clinical information of each participant.
- Line 153: The rationale for why the test order could avoid fatigue was not sufficiently explained. For example, was the order arranged from low-intensity to high-intensity tests? Response: The information was added in section methods, line174.
- The effect of rural sample exclusion on the representativeness of the results should be explained more fully. Response: Transportation within the Amazon is mainly by boat. Rural communities are located deep within the Amazon rainforest and are very difficult to access, both for researchers and residents. For this reason, rural inhabitants were not included in the assessments. We also sought to avoid bias in the assessments, as they are not representative of the population living in the interior of the state of Amazonas. For this reason, we mention in the methods section: “Elderly people living in rural areas were excluded from the study due to difficulties in accessing the assessment location (distance and necessary means of transport). For a better understanding of the geographical location of the urban areas where the assessments were carried out, the GPS locations are indicated in the methods section.
- Both the abstract and results mention that females performed better in certain aspects, but the reasons for gender differences are not sufficiently discussed. Please provide additional elaboration. Response: The information was added
- The discussion of intervention recommendations is too general and should be more specific and consider feasibility. Response: The information was added
- The discussion section should be logically structured with clear and natural paragraph divisions. Response: The information was structured

Reviewer 2 Report
Comments and Suggestions for Authors
Thank you for the opportunity to review the manuscript entitled “Functional Fitness of Low-income Community-Dwelling Older Adults in Amazonian Brazil.”
- Title: Please include the date of the study.
- Abstract Background: Begin with the importance of functional fitness.
- Abstract: In the methods section, provide the date of the study, data analysis software used, and the sampling technique.
- Terminology: Use "older adults" or "older people" instead of "elderly."
- Introduction: Include demographic information and health status of older adults in Amazonian Brazil.
- Sampling Technique: Provide detailed information about the sampling technique used.
- Data Collection: Explain how data collection was conducted.
- Results Section: Data analysis requires advanced inferential statistical tests.
- Practical Implications: Provide practical implications of the study.
- Limitations Section: Mention the exclusion of older adults living in rural areas.
Author Response
Dear Reviewer,
On behalf of the manuscript authors, we appreciate all comments from the reviewers. The authors believe that the comments have significantly improved the quality of the manuscript. For better analysis by the reviewers, we have responded individually to all comments.
Available for further information.
Kind Regards
Duarte Henriques-Neto
Reviewer 2
Thank you for the opportunity to review the manuscript entitled “Functional Fitness of Low-income Community-Dwelling Older Adults in Amazonian Brazil.”
- Title: Please include the date of the study. Response: The authors did not add the date to the title to avoid misinterpretation by readers and data. The reason for this is that this database was built over a period of seven years, as the research team spent about a year in each municipality. As the reviewer can see, the municipalities are geographically isolated, and it is impossible to collect data in a short period of time. For this reason, we opted to conduct a cross-sectional study to try to identify the functional profile of the population living in isolation in this state.
- Abstract Background: Begin with the importance of functional fitness. Response: The information was added – Lines 18-19
- Abstract: In the methods section, provide the date of the study, the data analysis software used, and the sampling technique. Response: To comply with the journal's rules, we explained the statistical analysis of the data in detail in the methods section of the manuscript.
- Terminology: Use "older adults" or "older people" instead of "elderly. Response: The term was changed.
- Introduction: Include demographic information and health status of older adults in Amazonian Brazil. Response: Demographic information was added in the section methods. To the best of our knowledge, there are no official or scientifically published data on the health status of this population other than our own data (i.e:https://doi.org/10.1093/geroni/igx004.1401 ; https://revistaretos.org/index.php/retos/article/view/78549/61697; https://journals.plos.org/plosone/article?id=10.1371/journal.pone.0320079). Our study is pioneering in this population with high limitations in accessing basic healthcare.
- Sampling Technique: Provide detailed information about the sampling technique used. Response: No sample calculation was performed, as there are no official data recorded for the entire population. For this reason, we included in the study limitations that further assessments are necessary for the data to be representative.
- Data Collection: Explain how data collection was conducted. Response: The information was added. Line 109-113.
- Results Section: Data analysis requires advanced inferential statistical tests.
Response: The inferential statistical and their results were added.
- Practical Implications: Provide practical implications of the study. Response: The information was added.
- Limitations Section: Mention the exclusion of older adults living in rural areas. Response: The information was added.

Reviewer 3 Report
Comments and Suggestions for Authors
First of all, I would like to thank both the editor and the authors for the opportunity to review this manuscript. There are several limitations that I believe should be addressed and clarified.
Firstly, abbreviations should be presented in full the first time they appear in the text, followed by the abbreviation in parentheses, and thereafter used only in abbreviated form (e.g., functional fitness [FF]). Is the study registered in any clinical trial database, such as ClinicalTrials.gov? What is meant by 'UEA premises'?
If the objective is to evaluate functional fitness in the older adult population, why were only participants without chronic diseases or conditions included? What were the exclusion criteria? Please ensure that the six-minute walk test (6MWT) is introduced with its full name followed by the abbreviation the first time it appears.
The first paragraph of the discussion should provide a brief summary of the study’s main findings. Additionally, please review the reference list, as it appears that different citation styles have been used within the same manuscript.
Author Response
Dear Reviewer,
On behalf of the manuscript authors, we appreciate all comments from the reviewers. The authors believe that the comments have significantly improved the quality of the manuscript. For better analysis by the reviewers, we have responded individually to all comments.
Available for further information.
Kind Regards
Duarte Henriques-Neto
Reviewer 3
First of all, I would like to thank both the editor and the authors for the opportunity to review this manuscript. There are several limitations that I believe should be addressed and clarified.
Firstly, abbreviations should be presented in full the first time they appear in the text, followed by the abbreviation in parentheses, and thereafter used only in abbreviated form (e.g., functional fitness [FF]). Is the study registered in any clinical trial database, such as ClinicalTrials.gov? What is meant by 'UEA premises'? Response: To avoid misinterpretations, we rewritten the paragraph “After explanations about the study procedures and risks, all participants signed the free and informed consent form. All assessments were carried out on previously approved ethical principles of the UEA”
If the objective is to evaluate functional fitness in the older adult population, why were only participants without chronic diseases or conditions included? What were the exclusion criteria? Please ensure that the six-minute walk test (6MWT) is introduced with its full name followed by the abbreviation the first time it appears. Response: The information was added.
The first paragraph of the discussion should provide a brief summary of the study’s main findings. Additionally, please review the reference list, as it appears that different citation styles have been used within the same manuscript. Response: The information was added and revised.
